# Design principles for sodium superionic conductors

Shuo Wang[1,5], Jiamin Fu[2,3,5], Yunsheng Liu[1], Ramanuja Srinivasan Saravanan[1], Jing Luo[2], Sixu Deng[2], Tsun-Kong Sham [3], Xueliang Sun [2] ✉ & Yifei Mo [1,4] ✉

Motivated by the high-performance solid-state lithium batteries enabled by lithium superionic conductors, sodium superionic conductor materials have great potential to empower sodium batteries with high energy, low cost, and sustainability. A critical challenge lies in designing and discovering sodium superionic conductors with high ionic conductivities to enable the development of solid-state sodium batteries. Here, by studying the structures and diffusion mechanisms of Li-ion versus Na-ion conducting solids, we reveal the structural feature of face-sharing high-coordination sites for fast sodium-ion conductors. By applying this feature as a design principle, we discover a number of Na-ion conductors in oxides, sulfides, and halides. Notably, we discover a chloride-based family of Na-ion conductors $Na_xM_yCl_6$ (M = La−Sm) with $UCl_3$-type structure and experimentally validate with the highest reported ionic conductivity. Our findings not only pave the way for the future development of sodium-ion conductors for sodium batteries, but also consolidate design principles of fast ion-conducting materials for a variety of energy applications.

Superionic conductor (SIC) is a unique type of materials exhibiting exceptionally high ionic conductivities, serving as critical components in a wide range of devices for energy storage and conversion, including solid-state batteries, solid-oxide fuel cells, solid-oxide electrolyzers, and ceramic membranes[1–4]. Among them, the lithium SICs, which replace liquid electrolytes in lithium-ion batteries as solid electrolytes, enable the next-generation solid-state lithium batteries with improved safety, high energy density, and long cycle life[5–7]. Thanks to the abundance and low cost of sodium resources, sodium SICs also attract great interest as solid electrolytes for solid-state sodium batteries[8–10] and sodium-sulfur batteries[11–13], which are economical and sustainable alternatives to current lithium-ion batteries. However, only a few materials are known as SICs, impeding the further development of these novel energy technologies. A long-standing challenge in materials science is how to rationally design and discover SIC materials with high ionic conductivities.

While Na-SICs beta-alumina[14] and NASICON[15] achieved high Na ionic conductivities back in 1970s, there are many discoveries of Li-SIC families in the past two decades (Supplementary Figs. 1 and 2), including Li-garnet $Li_7La_3Zr_2O_{12}$ (LLZO) oxides[16], $Li_{10}GeP_2S_{12}$ (LGPS)[1], $Li_7P_3S_{11}$ sulfides[17], Li-argyrodite $Li_6PS_5X$ (X = Cl, Br, I)[18], $Li_3MX_6$ (M = Y, Sc, In, Er, etc., X = Cl, Br) halides[19–21], with high Li ionic conductivities $\sigma_{RT}$ on the order of 1–10 mS/cm at room temperature (RT). By understanding the $Li^+$ diffusion mechanisms in these SICs[22–24], scientists established multiple design principles for Li-SICs, for example, based on body-centered cubic (bcc) anion framework[25], concerted migration[26], and corner-sharing crystal structural framework[27]. By successfully employing these design principles, many Li-SICs, including $LiZnPS_4$, $LiTaSiO_5$, $LiGa(SeO_3)_2$, were discovered from first principles computation and experimentally verified[27–29].

Despite the rapid advancement in the field of Li-SICs, the development and discovery of Na-SICs have been greatly lagging. Given the

[1]Department of Materials Science and Engineering, University of Maryland, College Park, MD 20742, USA. [2]Department of Mechanical and Materials Engineering, University of Western Ontario, London, ON N6A 5B9, Canada. [3]Department of Chemistry, University of Western Ontario, London, ON N6A 5B7, Canada. [4]Maryland Energy Innovation Institute, University of Maryland, College Park, MD 20742, USA. [5]These authors contributed equally: Shuo Wang, Jiamin Fu. ✉e-mail: xsun9@uwo.ca; yfmo@umd.edu

chemical similarities between Li$^+$ and Na$^+$, a common approach of developing Na-SICs is to make Na counterparts of known Li SICs, but the outcomes are underwhelming. For example, while Li-SIC LGPS ($\sigma_{RT}$ = 12 mS/cm) is a major breakthrough[1], its Na-analogy Na$_{10}$SnP$_2$S$_{12}$ in the same structure only achieved a much lower ionic conductivity $\sigma_{RT}$ of 0.4 mS/cm.[30] For another inspiring discovery of halide Li-SIC Li$_3$MX$_6$ (M = Y, Sc, In, Er, etc., X = Cl, Br) with high ionic conductivity $\sigma_{RT}$ > 1 mS/cm,[19–21] the Na-halide counterpart Na$_2$ZrCl$_6$ with the same hexagonal close-packed (hcp) Cl-anion framework as Li$_3$YCl$_6$ only exhibits a limited $\sigma_{RT}$ of 0.02 mS/cm.[31] Moreover, Li-garnet[16] and Li-argyrodite[18,22], which are successfully demonstrated as solid electro-lytes for solid-state lithium batteries with excellent cell performances[32,33], have no Na counterparts. For Na SICs, W-doped Na$_3$SbS$_4$[34] has a reported $\sigma_{RT}$ of 32 mS/cm, but its structures differ significantly from its Li-counterparts. As clearly indicated by these facts, the crystal structures that yield the low energy barriers for Li$^+$ and Na$^+$ diffusion are different, but are not yet understood. It is not clear how the knowledge derived from known Na-SICs can be utilized to design and discover more Na-SICs. One cannot simply duplicate the design principles for Li-ion conductors to discover Na-ion conductors.

Given many recent discoveries in Li-SICs with $\sigma_{RT}$ higher than 1–10 mS/cm, the discovery of Na-SICs is an under-explored opportu-nity. In comparison to Li-SICs, no oxide structures of Na-SICs with $\sigma_{RT}$ > 1 mS/cm was discovered since 1970s (Supplementary Table 1), and the halide Na-SICs still show $\sigma_{RT}$ two orders of magnitude lower than Li-halides[31]. While Li-SICs empower solid-state lithium batteries with ever-improving cell performances, the limited availability of Na-SICs has been impeding the development and innovation of sodium batteries. Thus the design principles applicable to Na-SICs are urgently needed.

In this study, we first reveal the fundamental differences between the crystal structures and diffusion mechanisms of Li$^+$ versus Na$^+$ by analyzing Li- and Na-conducting oxides, sulfides, and halides. Through this understanding, a unique feature of fast Na-ion conductors is identified, and is then formulated as a design principle. Applying our design principle in a high-throughput computational screening, we discover over a dozen of structural families of Na-ion conductors with high ionic conductivities. In particular, a halide family of Na-ion conductors Na$_x$M$_y$Cl$_6$ (M = La–Sm), is discovered and successfully synthesized with $\sigma_{RT}$ of 1.4 mS/cm, the highest among sodium halides.

## Results

### Different site preferences for Na$^+$ and Li$^+$ in crystal structures

We first analyze and compare the structures of representative Li$^+$ and Na$^+$ SICs (Fig. 1), and identify the differences in the local geometry of Li$^+$/Na$^+$ sites in forming the diffusion channels. There are significant differences between the Li$^+$/Na$^+$ site coordination number (CN), which is the number of the nearest neighbor anions (Methods). Whereas Li$^+$ occupies and migrates among tetrahedral (Tet) sites in the LGPS Li-SIC, Na$^+$ occupies sites with higher CNs of ≥ 5 in Na-SICs, such as beta-alumina (CN = 6–8), NASICON (CN = 5, 6, 8), Na$_3$SbS$_4$ (CN = 8) (Fig. 1), and Na$_3$PS$_4$ (CN = 6) (Supplementary Fig. 3). The preferences of Na$^+$ for higher CN can be understood by the larger Na$^+$ radius ($r_{Na+}$ = 1.02 Å) compared to Li$^+$ ($r_{Li+}$ = 0.76 Å) according to Pauling's rules[35]. Among all Na- and Li-containing oxides, sulfides, and chlorides, the preference for high-CN Na$^+$ sites is general as confirmed by our analyses (Sup-plementary Fig. 4).

### Why the design principles for Li-ion conductors cannot be duplicated for Na-ion conductors?

Here we illustrate the design principles for Li-ion conductors cannot be duplicated for Na-ion conductors, because of the different preferred site-coordination of Na$^+$ and Li$^+$. We investigate and simulate the energy barrier of Li$^+$ and Na$^+$ migration in the model systems with fixed bcc, hcp, and face-centered cubic (fcc) anion sublattices with no cation (Fig. 2a–c). In a bcc anion framework, such as in LGPS and Li$_7$P$_3$S$_{11}$, Li$^+$ sits at tetrahedral (Tet) sites and migrates by crossing an anion-triangle bottleneck to another face-sharing Tet site, giving a low energy barrier of 0.12 eV for Li$^+$ migration (Fig. 2a). Proposed and demonstrated by Wang et al.[25], the bcc anion framework as a design principle for Li-SICs has successfully led to the discovery of a Li-SIC LiZnPS$_4$[29]. However, in Na-containing compounds, because of the strong preference of high-CN Na$^+$ sites (Supplementary Fig. 4), it would be difficult to form per-colation diffusion channels solely by Tet Na$^+$ sites, as desired in the bcc-anion-framework design principle. There has been no Na-containing material with the bcc-anion framework.

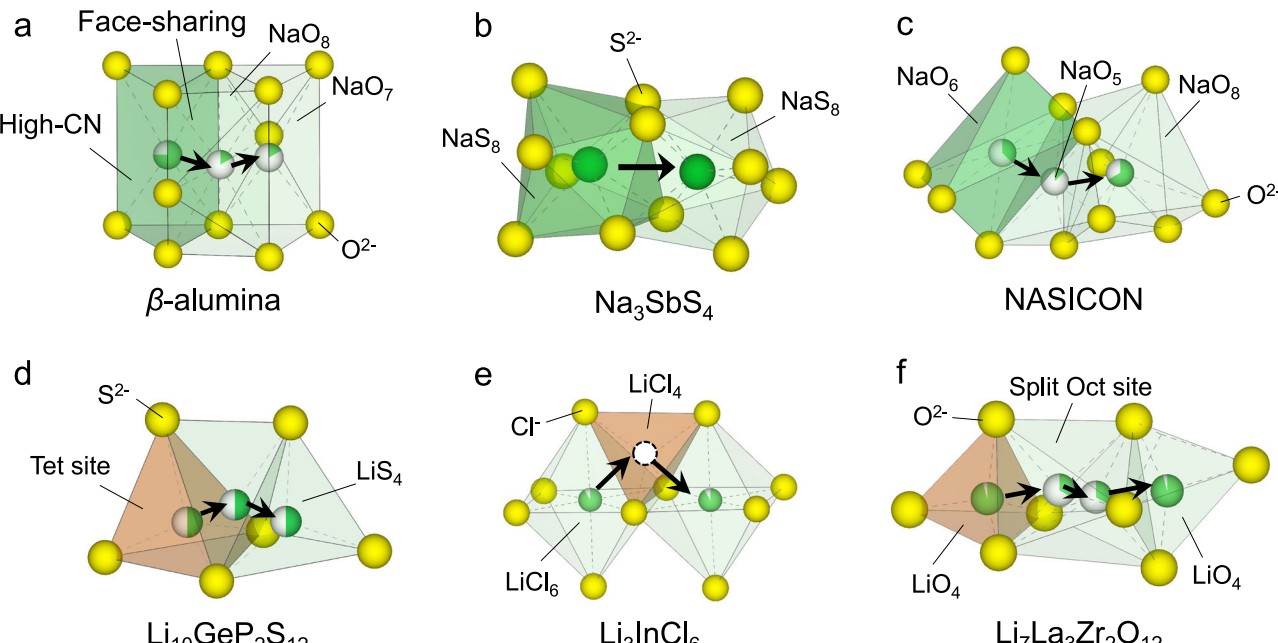

**Fig. 1 | The ion diffusion channel in Li/Na-ion conductors.** The Li$^+$/Na$^+$ sites (green) coordinated with O$^{2-}$/S$^{2-}$/Cl$^-$ anions (yellow) connected to form the diffusion channel in representative **a–c** sodium and **d–f** lithium SICs.

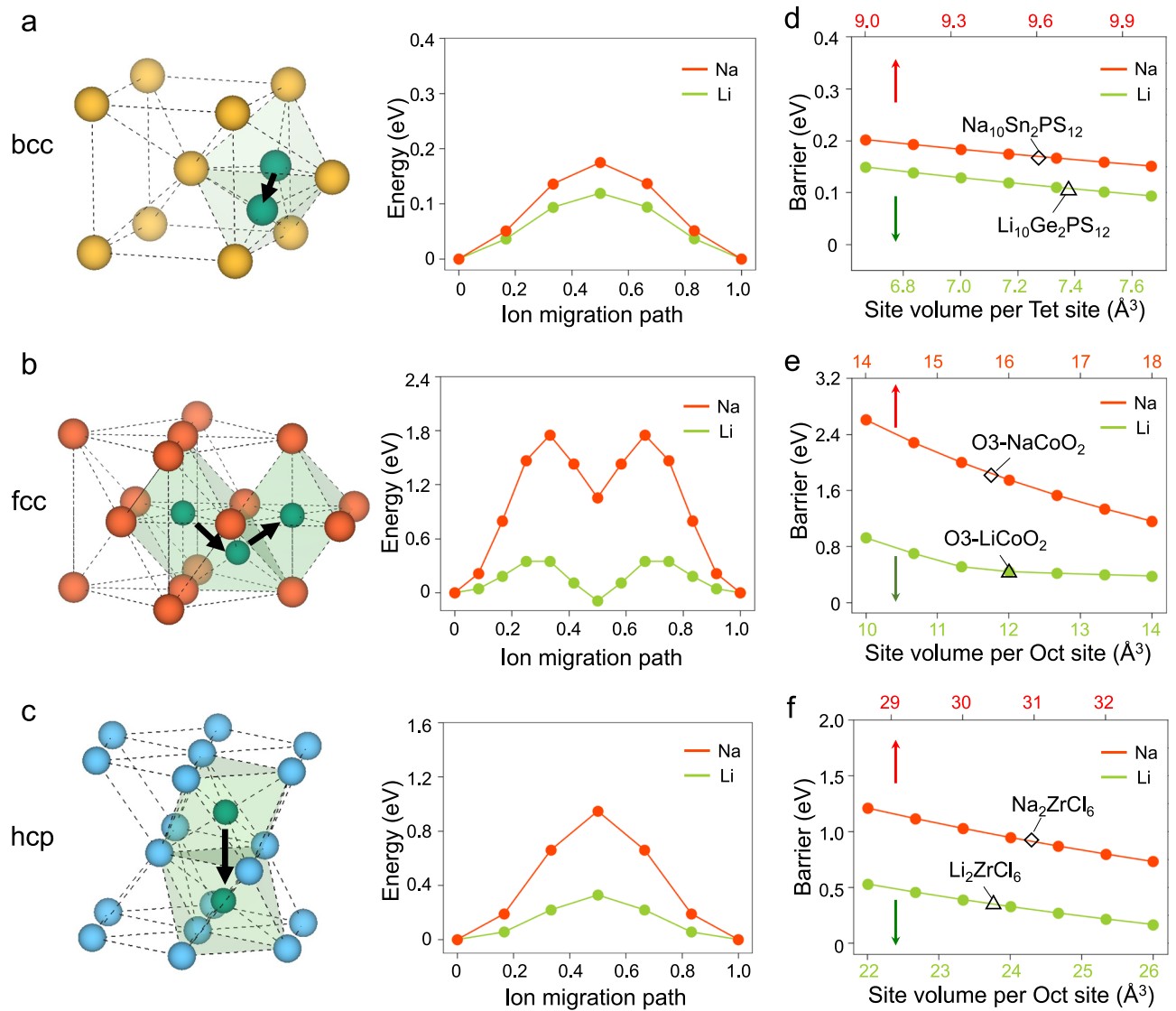

**Fig. 2 | Lithium-ion and sodium-ion diffusion in model anion sublattices.** **a**–**c** The diffusion pathways (left) and corresponding energy profile (right) for single Li$^+$ (green) and Na$^+$ (red) migration in fixed **a** body-centered cubic (bcc) S$^{2-}$, **b** face-centered cubic (fcc) O$^{2-}$, **c** hexagonal close-packed (hcp) Cl$^-$ anion sublattice. The fixed anion lattice is set to have the octahedral (Oct) and tetrahedral (Tet) site volume as the real materials, O3-type LiCoO$_2$ (Oct: 12.0 Å$^3$) and NaCoO$_2$ (Oct: 15.7 Å$^3$) for fcc O$^{2-}$, Li$_2$ZrCl$_6$ (Oct: 23.8 Å$^3$) and Na$_2$ZrCl$_6$ (Oct: 30.9 Å$^3$) for hcp Cl$^-$, Li$_{10}$GeP$_2$S$_{12}$ (Tet: 7.4 Å$^3$) and Na$_{10}$SnP$_2$S$_{12}$ (Tet: 9.6 Å$^3$) for bcc S$^{2-}$. **d**–**f** The energy barrier of Na$^+$ (red) and Li$^+$ (green) migration as a function of site volume in the fixed **d** bcc S$^{2-}$, **e** fcc O$^{2-}$, **f** hcp Cl$^-$ anion sublattice.

For the close-packed fcc and hcp anion sublattices, as commonly found in Li-chlorides and bromides SICs Li$_3$MX$_6$, Li$^+$ migration has low energy barriers of 0.2–0.3 eV (Supplementary Figs. 7 and 8), which are sufficiently low for fast Li-ion conductors[36]. The Na$^+$-conducting O3-type NaMO$_2$ (M = Co, Mn, Ni, etc.) also has an fcc anion sublattice, in which Na$^+$ occupies octahedral (Oct) sites (CN = 6) and migrates between Oct sites through intermediate Tet site (CN = 4). The intermediate Tet-site has a higher site energy than the Oct sites, as the high CN preference of Na$^+$ makes Tet-sites unfavorable, thus giving a high migration barrier of >1.0 eV along the Oct-Tet-Oct pathway (Fig. 2b and Supplementary Figs. 7 and 8). In the hcp anion sublattice, as in Na$_2$ZrCl$_6$, this Oct-Tet-Oct pathway is also required for 3D conduction. In 1D Oct-Oct pathways along the $c$-axis, Na$^+$ migrates among face-sharing Oct sites across an anion-triangle bottleneck (Fig. 2c). This anion-triangle bottleneck has an appropriate size for Li$^+$ migration but is too small for Na$^+$. As a result, the hcp Cl$^-$ anion sublattice gives a low Li$^+$ migration barrier of ~0.3 eV in Li$_3$MX$_6$ halide Li-SICs, but gives a high Na$^+$ migration barrier of ~0.9 eV (Fig. 2c). This explains the limited Na-ion conduction in

Na$_2$ZrCl$_6$ with the same hcp Cl-anion framework as Li$_3$YCl$_6$, as reported in experiments[31].

In summary, low-barrier Na$^+$ migration is difficult to be realized in these typical materials structures given by the design principles for Li-ion conductors (Fig. 2d–f). While the mechanisms, such as concerted migration[23,26], poly-anion rotation[37,38], and phonon lattice effect[39,40], may further facilitate ion diffusion (see section "High-throughput discovery for Na-ion conductors"), having the structural framework with the flat energy landscape is the first requirement to have fast ion conductors. Compared to Li$^+$, Na$^+$ diffusion in solid crystal structures have two critical limitations. (1) Na$^+$ is unfavorable in the Tet (low-CN) sites, but typical structures give migration pathways with intermediate Tet sites, causing a high migration-energy barrier (Figs. 2b and 3). (2) Na$^+$ requires a larger bottleneck size of the diffusion channel. As quantified in our analyses on the percolation radius $p_r$, which is the maximum radius of a sphere that can percolate the structure across at least one dimension (Methods), Na SICs have much larger percolation radii ($p_r$ = 0.88–1.28 Å) than Li SICs ($p_r$ = 0.52–0.72 Å) (Supplementary Fig. 4 and Supplementary Table 2). As shown above (Fig. 2), typical

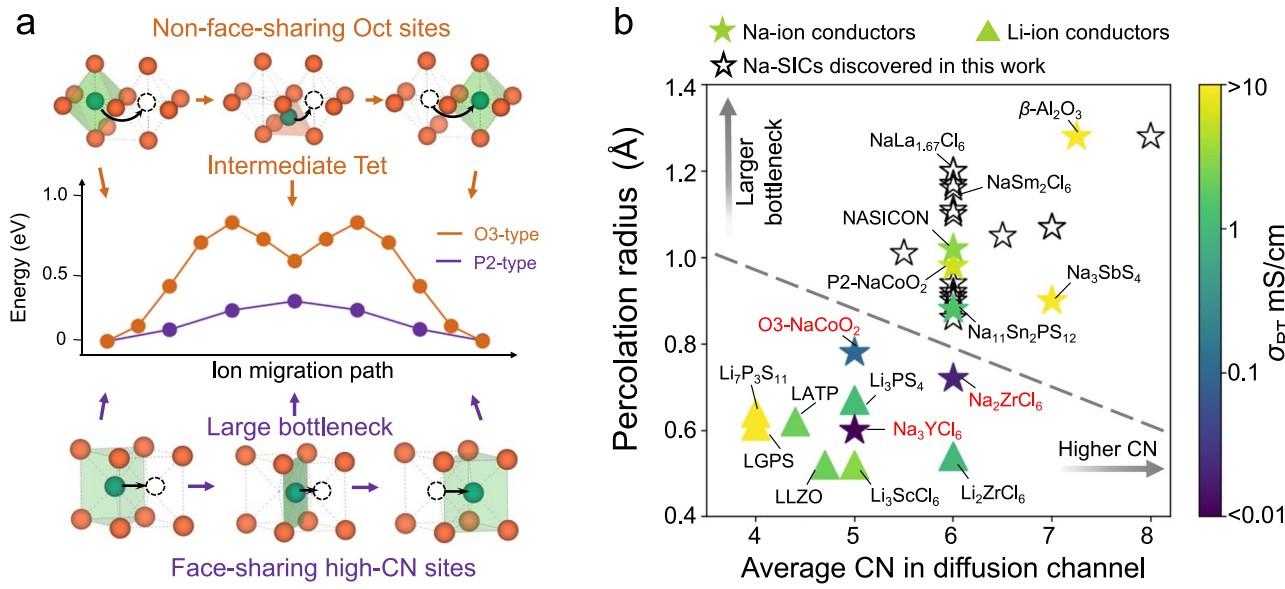

**Fig. 3 | Design principles for sodium-ion conductors. a** The schematics illustrate (lower) the Prism-Prism pathway between face-sharing high-CN sites in P2-type $NaMO_2$, in comparison to (upper) the Oct-Tet-Oct pathway with intermediate Tet site in O3-type $NaMO_2$, with (middle) the calculated energy profile for $Na^+$ migration in the fixed O-anion sublattice at the lattice volume of 19.3 $Å^3$ per $O^{2-}$. **b** The comparison of the percolation radii $p_r$ and the average CNs of Li/Na sites along the diffusion channels in Li-ion and Na-ion conductors showing a clear distinction.

crystal structures cannot offer such large bottlenecks for low-barrier Na-ion diffusion. How do the structures of Na-SICs overcome these limitations to achieve fast Na-ion conduction?

**Design principles for Na-ion conductors**

Here we identify and propose the feature of face-sharing high-CN sites for fast Na-ion conductors. Specifically, the $Na^+$ diffusion channel should be comprised of only high-CN sites (CN ≥ 5) that are face-sharing with a large bottleneck size to form percolation in the crystal structure. While high-CN $Na^+$ sites are general (Supplementary Fig. 4), having these high-CN sites be face-sharing is unique among crystal structures (Fig. 1 and Supplementary Fig. 3). For example, in Na beta-alumina (Fig. 1), the $Na^+$ occupies the trigonal prismatic (CN = 6) or capped trigonal prismatic sites (CN = 7, 8), which are connected via face-sharing $O^{2-}$-rectangle with a large bottleneck size ($p_r$ = 1.28 Å). Similarly, the P2-type $NaCoO_2$, (Figs. 1 and 3a) which is reported to have high RT ionic conductivity up to 6 mS/cm,[41] also has a diffusion network of equivalent prismatic (Prism) Na sites connected by face-sharing $O^{2-}$-rectangle with a large bottleneck size ($p_r$ = 0.92 Å) (Fig. 3a). The Prism-Prism pathway has a low energy barrier of 0.26 eV for $Na^+$ migration in the fixed $O^{2-}$ anion sublattice model of the P2-type $NaMO_2$ (Fig. 3a). By contrast, for the structures with the high-CN sites not face-sharing, e.g. the fcc anion sublattice as in O3-type $NaMO_2$, there is unfavorable Tet site as intermediate along the diffusion pathway, causing a high energy barrier of 0.76 eV at the same lattice volume of P2 (Fig. 3a). Face-sharing high-CN sites allow a direct $Na^+$ hopping among equivalent high-CN sites with a large bottleneck size and without an unfavorable intermediate Tet site (Fig. 3a), hence overcoming the aforementioned limitations of $Na^+$ diffusion.

In Fig. 3b, we compare Na-ion and Li-ion conductors for their average CN of $Li^+$/$Na^+$ sites along the diffusion channels and their bottleneck sizes ($p_r$). The known Na-SICs, such as NASICON, beta-alumina, and $Na_3SbS_4$, have high average CNs ≥ 6 along their diffusion channels in their structures and also have large bottleneck size ($p_r$ ≥ 0.88 Å) (Fig. 3b). By contrast, other Na-containing materials that do not exhibit face-sharing high-CN sites and large bottleneck size show lower conductivity. For example, O3-type $NaMO_2$ structure and $Na_3YCl_6$ have lower average CN due to the intermediate Tet-site along

their diffusion channels, and the $Na_2ZrCl_6$ with hcp anion framework has a small bottleneck size ($p_r$ = 0.73 Å). By contrast, all Li-SICs have low average CNs (Fig. 3b). For example, LGPS and $Li_7P_3S_{11}$ with face-sharing Tet sites (average CN = 4) (Figs. 1 and 3b) show the highest ionic conductivity, according to the bcc-anion-framework design principle[25]. Other non-bcc Li-SICs, such as LLZO and $Li_3ScCl_6$, have higher average CN because of the non-Tet sites along the diffusion pathways, and in general show ionic conductivities not as high as those with bcc-anion framework (Fig. 3b). This comparison chart (Fig. 3b) clearly demonstrates the key feature of face-sharing high-CN sites in Na-SICs highly distinct from Li-SICs.

**High-throughput discovery for Na-ion conductors**

Here, our design principle is employed in a high-throughput computation screening to discover Na SICs (Fig. 4a), among Na-containing oxides, sulfides, and chlorides in the Inorganic Crystal Structure Database (ICSD)[42]. In addition to basic checks and practical considerations of materials (Methods), two screening criteria following our design principle were employed: (1) the diffusion channel consists of only high-CN sites (CN ≥ 5) that are face-sharing and are connected within a distance of 3.1 Å for oxides and 3.5 Å for sulfides/chlorides; (2) a large percolation radius $p_r$ > 0.85 Å for sulfides/chlorides and $p_r$ > 0.90 Å for oxides. Using these screening criteria, all known sodium SICs, including NASICON, beta-alumina, $Na_3SbS_4$, $Na_3PS_4$, and $Na_{11}Sn_2PS_{12}$, are identified (Supplementary Table 3), and 35 unique structures are discovered as candidates of Na-SICs. These candidate structures are further studied by aliovalent substitution to tune $Na^+$ content, in order to enhance ionic conductivity through different mobile-ion concentration[24,28,29] or activating concerted migration mechanisms[23,29,43] ("Methods"). Those substituted materials with good stability (energy above hull <100 meV/atom) are then evaluated for ionic conductivity using ab initio molecular dynamics (AIMD) simulations. Other mechanisms that may facilitate ion migration, such as concerted migration[26,37], cooperative polyanion rotation[37,38], and phonon effects[39,40], if occur in the corresponding materials, would be captured in the AIMD simulations. Among them, 19 Na-SICs with $\sigma_{RT}$ > 0.1 mS/cm are discovered (Table 1). Given the large statistical variances of diffusivities and the extrapolation of the Arrhenius

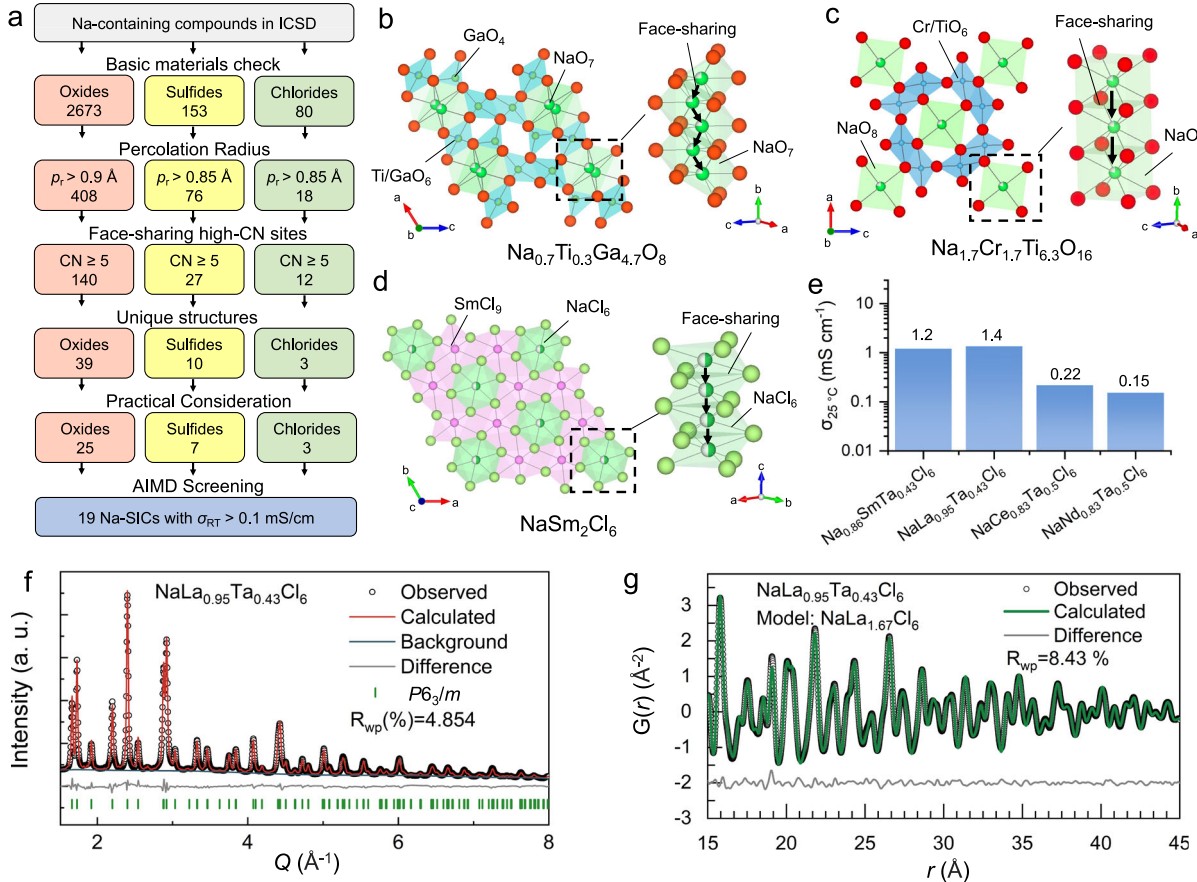

**Fig. 4 | Discovery of sodium superionic conductors. a** High-throughput screening of Na-containing oxides, sulfides, and chlorides. **b–d** The crystal structures of representative discovered sodium SICs and (inset) their diffusion channels consist of face-sharing high-CN sites. **e** The high experimental Na$^+$ conductivities $\sigma_{RT}$ of

Na$_{0.86}$SmTa$_{0.43}$Cl$_6$, NaLa$_{0.95}$Ta$_{0.43}$Cl$_6$, NaCe$_{0.83}$Ta$_{0.5}$Cl$_6$, and NaNd$_{0.83}$Ta$_{0.5}$Cl$_6$. **f** The Rietveld refinements of synchrotron-based diffraction pattern and **g** the fitting result of the pair distribution function of NaLa$_{0.95}$Ta$_{0.43}$Cl$_6$.

relations, the results of AIMD simulations should only be interpreted as the confirmation of these structure frameworks are fast Na-ion conducting if an appropriate level of aliovalent doping can be achieved. All these structures of discovered Na-SICs show the unique feature of face-sharing high-CN sites (Fig. 4b–d), confirming our design principle.

Among the discovered oxides Na-SICs, Na$_{0.67}$Ti$_{0.33}$Ga$_{4.67}$O$_8$ (doped from ICSD-34196), from the tunneled alkali titan-gallate family[44], has a low activation barrier ($E_a$ = 0.14 eV) and a high conductivity ($\sigma_{RT}$ = 8.8 mS/cm) in AIMD simulations (Supplementary Fig. 12). Its crystal structural framework consists of connected GaO$_4$ tetrahedra and GaO$_6$/TiO$_6$ (Fig. 4b), and Na$^+$ occupies capped triagonal prismatic sites (CN = 7), which are face-sharing with O-rectangle bottlenecks with a large bottleneck ($p_r$ = 1.07 Å). In another discovered Na-SIC Na$_{1.33}$Mg$_{0.67}$Ti$_{7.33}$O$_{16}$ (doped from ICSD-50764)[45], Na$^+$ migrates between equivalent eight-coordinated sites through a large O-rectangle bottleneck ($p_r$ = 1.28 Å) (Fig. 4c), resulting in fast 1D diffusion ($E_a$ = 0.30 eV and $\sigma_{RT}$ = 1.7 mS/cm). While the fast ion conduction is confirmed in the bulk phases of these materials, those materials that have 1D fast diffusion channels may be susceptible to the blocking effect of defects and grain boundaries[5,46,47], which deserve future studies.

A halide family Na$_x$M$_y$Cl$_6$ with UCl$_3$-type structure[48] with a wide range of compositions (M = lanthanides, $x$ = 0–1, $y$ = 1.67–2) is discovered as Na-SICs. In contrast to Li$_3$MX$_6$ (M = Y, Sc, In, Er, etc., X = Cl, Br) halide Li-SICs with closed-packed anion framework, the structure of Na$_x$M$_y$Cl$_6$ exhibits face-sharing octahedral Na$^+$ sites forming 1D diffusion channel along $c$-axis (Fig. 4d). These octahedral sites are distorted, thus enlarge the Cl-triangle bottleneck ($p_r$ = 1.16 Å). We evaluate NaSm$_2$Cl$_6$ and NaLa$_{1.67}$Cl$_6$ for Na$^+$ diffusion. In AIMD

simulations, both exhibited fast Na$^+$ conduction along the 1D channels with a low activation barrier of 0.13 eV and 0.15 eV, respectively (Table 1 and Supplementary Fig. 14). Experimentally, we successfully synthesized this family of Na$_{3x}$M$_{2-x}$Cl$_6$-containing (M = La, Ce, Nd, Sm) halide Na$^+$ conductors using ball-milling methods (Methods), and acquired high ionic conductivities of 1.2, 1.4, 0.22, 0.15 mS/cm at 25 °C for Na$_{0.86}$SmTa$_{0.43}$Cl$_6$, NaLa$_{0.95}$Ta$_{0.43}$Cl$_6$, NaCe$_{0.83}$Ta$_{0.5}$Cl$_6$, and NaNd$_{0.83}$Ta$_{0.5}$Cl$_6$, respectively (Fig. 4e). The X-ray diffraction pattern (XRD) verified the UCl$_3$-type structure as the dominant phase (Supplementary Fig. 16). The synchrotron-based XRD refinement results and pair distribution function of NaLa$_{0.95}$Ta$_{0.43}$Cl$_6$ (Fig. 4f, g and Supplementary Tables 12 and 13) confirmed the dominant crystalline phase NaLa$_{1.67}$Cl$_6$, and additionally NaTaCl$_6$. The NaTaCl$_6$ is formed as secondary phase in the inter-grain, and is known to have relatively lower ionic conductivity ($\sigma_{RT}$ = 0.045 mS/cm). The NaLa$_{0.95}$Ta$_{0.43}$Cl$_6$ composite of NaLa$_{1.67}$Cl$_6$ and NaTaCl$_6$ achieved the highest reported $\sigma_{RT}$ of 1.4 mS/cm (Fig. 4e and Supplementary Fig. 17), a significant improvement over the previous halide Na-ion conductor Na$_2$ZrCl$_6$ with $\sigma_{RT}$ of 0.02 mS/cm[31]. Given that the secondary phase NaTaCl$_6$ has a relatively lower ionic conductivity, the bulk phase of NaLa$_{1.67}$Cl$_6$ should have high ionic conductivity, in good agreement with the AIMD simulations (Supplementary Fig. 17). This discovery of a chloride Na-SIC family with the highest reported $\sigma_{RT}$ is a strong validation of our design principle for fast Na-ion conductors.

## Discussion
As the key underlying mechanism of our design principle, the unique feature of face-sharing high-CN sites gives direct ion-migration

**Table 1 | Sodium superionic conductors predicted by AIMD simulations**

| ICSD-IDs | Original composition | Origin/dopant | Doped composition | $E_{hull}$ (meV/atom) | $E_a$ (eV) | $\sigma$ at 300 K (mS/cm) | Error bound [$\sigma_{min}$, $\sigma_{max}$] (mS/cm) |
|---|---|---|---|---|---|---|---|
| 78919 | $Na_7Y_2P_7O_{24}$ | $P^{5+}/Mo^{6+}$ | $Na_{6.5}Y_2Mo_{0.5}P_{6.5}O_{24}$ | 23 | 0.23 ± 0.07 | 0.5 | [0.03, 9.4] |
| 50764 | $Na_{1.7}Cr_{1.7}Ti_{6.3}O_{16}$ | $Cr^{3+}/Ti^{4+}$, $Mg^{2+}$ | $Na_{1.33}Mg_{0.67}Ti_{7.33}O_{16}$ | 31 | 0.30 ± 0.04 | 1.7 | [0.3, 9.3] |
| 34196 | $Na_{0.7}Ti_{0.3}Ga_{4.7}O_8$ | $Ga^{3+}/Ti^{4+}$ | $Na_{0.67}Ti_{0.33}Ga_{4.67}O_8$ | 20 | 0.14 ± 0.05 | 8.8 | [1.5, 53] |
| 108816 | $NaTi_2Ga_5O_{12}$ | $Ga^{3+}/Ti^{4+}$ | $Na_{0.67}Ti_{2.33}Ga_{4.67}O_{12}$ | 10 | 0.26 ± 0.05 | 0.4 | [0.04, 3.3] |
| 155643 | $Na_{0.8}Ti_{1.2}Ga_{4.8}O_{10}$ | $Ga^{3+}/Ti^{4+}$ | $Na_{0.67}Ti_{1.33}Ga_{4.67}O_{10}$ | 26 | 0.26 ± 0.05 | 0.8 | [0.1, 5.0] |
| 163234 | $Na_2V_3O_7$ | $V^{4+}/V^{5+}$ | $Na_{1.33}V_3O_7$ | 40 | 0.34 ± 0.06 | 0.1 | [0.01, 1.4] |
| 262512 | $Na_3Nb_4As_3O_{19}$ | $O^{2-}/F^-$ | $Na_2Nb_4As_3O_{18}F$ | 43 | 0.24 ± 0.07 | 1.0 | [0.06, 17.9] |
| 39248 | $NaTiPO_5$ | – | – | 29 | 0.19 ± 0.06 | 4.3 | [0.3, 64] |
| 39788 | $NaGeSbO_5$ | $Ge^{4+}/P^{5+}$ | $Na_{0.75}Ge_{0.75}P_{0.25}SbO_5$ | 29 | 0.18 ± 0.05 | 7.2 | [0.7, 76] |
| 239705 | $Na_2ZnGe_2S_6$ | $Ge^{4+}/Zn^{2+}$ | $Na_{2.25}Zn_{1.125}Ge_{1.875}S_6$ | 20 | 0.25 ± 0.06 | 0.7 | [0.08, 6.9] |
| 300175 | $Na_5InS_4$ | $In^{3+}/Sn^{4+}$ | $Na_{4.5}In_{0.5}Sn_{0.5}S_4$ | 46 | 0.35 ± 0.04 | 0.2 | [0.03, 0.8] |
| 33236 | $Na_6ZnS_4$ | $Zn^{2+}/Ga^{3+}$ | $Na_{5.5}Zn_{0.5}Ga_{0.5}S_4$ | 23 | 0.32 ± 0.05 | 0.2 | [0.02, 1.8] |
| 62579 | $Na_5FeS_4$ | $Fe^{3+}/In^{3+}$, $Sn^{4+}$ | $Na_{4.75}In_{0.75}Sn_{0.25}S_4$ | 62 | 0.28 ± 0.04 | 1.9 | [0.4, 9.1] |
| 234888 | $Na_3ZnGaS_4$ | $Zn^{2+}/Ga^{3+}$ | $Na_{2.75}Zn_{0.75}Ga_{1.25}S_4$ | 42 | 0.25 ± 0.06 | 0.5 | [0.04, 6.2] |
| 200983 | $Na_{0.7}Ti_{0.3}Cr_{0.7}S_2$ | $Cr^{3+}/Ti^{4+}$ | $Na_{0.67}TiS_2$ | 3 | 0.23 ± 0.04 | 2.2 | [0.4, 14] |
| 74928 | $NaSm_2Cl_6$ | – | – | 53 | 0.13 ± 0.03 | 131 | [41 430] |
|  |  | $Sm^{2+}/La^{3+}$ | $NaLa_{1.67}Cl_6$ | 9 | 0.15 ± 0.03 | 56.5 | [13, 240] |
| 69343 | $Na_2MgCl_4$ | $Mg^{2+}/Er^{3+}$ | $Na_{1.33}Mg_{0.33}Er_{0.67}Cl_4$ | 44 | 0.35 ± 0.06 | 0.1 | [0.01, 1.3] |
| 11165 | $Na_2Ti_3Cl_8$ | $Ti^{2+}/Ti^{3+}$ | $Na_{1.67}Ti_3Cl_8$ | 0 | 0.30 ± 0.07 | 0.2 | [0.01, 3.5] |

pathways among equivalent, favorable $Na^+$ sites, with small CN changes and a large bottleneck, thus giving a low energy barrier (Fig. 3a). By contrast, in the structures with $Na^+$ sites that are not face-sharing, the diffusion pathways include intermediate Tet sites, which are unfavorable for $Na^+$ and thus cause high energy barriers (Figs. 2c and 3a). In comparison, in the Li-SICs with a bcc anion framework, the diffusion channels of face-sharing Tet-sites[25], which are generally favorable for $Li^+$, lead to the lowest energy barrier (Figs. 2a and 3b)[25]. The high-CN preference of $Na^+$ versus $Li^+$ explains why the structures of Li- and Na-SICs are different and why the optimal Li-SIC structures cannot be duplicated as Na-SICs (Figs. 1 and 3b). Regardless of the coordination preferences of the mobile-ions, the crystal structures of fast ion conductors should form direct migration pathways among equivalent favorable sites to minimize energy barrier. Here the design principles and desired features for fast $Li^+$ and $Na^+$ conductors are consolidated and generalized.

Furthermore, this generalized design principle can unify the understanding of Li-SICs with or without bcc-anion-framework. For example, in the LLZO garnet (Fig. 1), the distorted Oct-Li sites can be considered as a split into two Tet-Li sites and thus form face-sharing Tet-sites channel for $Li^+$ diffusion (Fig. 1), similar to that in LGPS, thus shows relatively flat potential energy landscape. By considering this distorted site splitting into multiple Tet sites, other non-bcc-anion-framework Li-SICs, such as LLZO, NASICON, $Li_6PS_5X$ (X = Cl, Br, I) (Fig. 1 and Supplementary Fig. 3), can be understood as face-sharing tetrahedral sites as in the bcc-anion-framework[21]. The mechanisms of site distortion are also attributed to the flattening of the energy landscape by raising site energies as reported in Li-SICs[26,27,43,49]. In addition, this splitting of a large local site volume into multiple equivalent sites of mobile-ions within a close distance causes the feature of enlarged $Li^+$ sites proposed by He et al.[34], which promotes the frustration and disordering of the overall $Li^+$ sublattice[43,50]. In this study, we find in the Na-SICs that site distortion has another beneficial effect, that is, enlarging the bottleneck size, which is critical for the diffusion of large-radius $Na^+$. For example, in NASCION, the $Na^+$ site is in the six-coordinated sites of distorted antiprism that form a face-sharing O-triangle bottleneck with a large bottleneck size of $p_r = 1.03$ Å. The triangle bottleneck size is enlarged by the distortion of Oct sites, in contrast to the non-distorted Oct-sites in the hcp anion sublattice (e.g., in $Na_2ZrCl_6$) which has a small bottleneck size ($p_r = 0.73$ Å). Therefore, the

distortion of mobile-ion sites is a key feature of Li-/Na-SICs, bringing multiple beneficial effects for fast ion diffusion.

Overall, these insights consolidate previous understandings on multiple features and mechanisms about the crystal structure frameworks of SICs. While this work focuses on the crystal structural frameworks that give low energy barriers for ion diffusion, more complex mechanisms e.g. concerted migration, cooperation motion, and phonon-assisted hopping, may be further added to devise more detailed design principles in the future. Furthermore, the effects of grain boundaries in addition to the bulk-phase fast ion conduction should be further studied in the future. As demonstrated in designing Na-SICs, these generalized design principles can be extended and applied for other types of fast ion conductors, facilitating the future design and discovery of SICs across vast materials space.

## Methods
### DFT calculation
All the calculations were carried out using Vienna Ab initio Simulation Package (VASP)[51] based on density functional theory (DFT) using Perdew–Burke–Ernzerhof (PBE)[52] generalized gradient approximation (GGA) described by the projector-augmented-wave (PAW) approach. The convergence parameters used in all static DFT calculations were set to be consistent with the Materials Project[53].

### Diffusion in fixed anion sublattice model
We performed the nudged elastic band (NEB) calculations to evaluate the migration of a single $Li^+/Na^+$ in a fixed bcc, fcc, and hcp anion sublattice of $O^{2-}$, $S^{2-}$, and $Cl^-$ with no other cations as in ref. 21. The anions were fixed, and the background charge was set to maintain the correct valence states of mobile-ion and anions (i.e. $Li^+$, $Na^+$, $O^{2-}$, $S^{2-}$, and $Cl^-$). The supercell models (54 atoms for bcc, 32 atoms for fcc, 36 atoms for hcp) and $\Gamma$-centered $2 \times 2 \times 2$ k-point grids were used. Static relaxation of mobile ion (i.e. $Li^+$ and $Na^+$) at initial and final sites within the fixed anion sublattice used an energy convergence criterion of $10^{-5}$ eV and a force convergence criterion of $10^{-2}$ eV/Å. A total of seven images interpolated between initial and end structures were used for the NEB calculations. In the NEB calculations, the anions were fixed, and the energy and force convergence criterion remained the same as the static relaxation of initial and final structures.

The lattice parameters of the bcc, fcc, and hcp anion model were set to have the same site volume as representative Li-SICs (i.e., O3-type LiCoO$_2$, Li$_2$ZrCl$_6$, Li$_{10}$GeP$_2$S$_{12}$) and Na-SICs (i.e., O3-type NaCoO$_2$, Na$_2$ZrCl$_6$, Na$_{10}$SnP$_2$S$_{12}$) as shown in Fig. 2a–c. The fixed anion sub-lattices extracted from real materials of P2- and O3-type NaCoO$_2$ are used in Fig. 3a. To consider the effects of changing volumes on Li$^+$/Na$^+$ migration, the models with a varying range of lattice parameters and lattice volume following the volume distribution of the Li/Na-containing compounds in the ICSD (Supplementary Fig. 6) were also conducted (Fig. 2d–f and Supplementary Figs. 7 and 8).

## Topological analysis of crystal structure frameworks

The topological analysis was performed to identify percolation radius, Na sites, site coordination, and diffusion network using in Zeo++[54] as in the previous study[43]. The topological analysis was performed to the crystal structural framework by removing all Li/Na ions from the crystal structures. The analyses were performed using the crystal ionic radii for each ion species in Pymatgen[55]. For sites with mixed partial occupancy of multiple cations, the smallest ionic radius of these cations was used. The percolation radius was calculated as the maximum radius of a sphere that can percolate the crystal structure across at least one direction.

The Na sites in the crystal structural framework were identified as follows. The structural framework was decomposed by the Voronoi–Dirichlet partition algorithm using the ions as the centers of the polyhedrons as implemented in Zeo++[54]. A Voronoi node was the vertices shared by the polyhedrons and corresponded to the center of a local void, which may be possible mobile-ion sites. These Voronoi nodes were further screened by the chemical environments suitable for Na$^+$ occupancy using the criteria based on the coulombic repulsion and bond valence (BV). Any nodes close to other non-Na cations were excluded, with the cutoff distance set to 2.6 Å for oxides/sulfides and 2.8 Å for chlorides. The BV was calculated for each Voronoi node, and those with BV in the range of 0.3–1.5 were kept. The distance cut-off and BV range were determined by analyzing Na-containing oxide, sulfide, and chlorides (Supplementary Figs. 10 and 11). Finally, the Na nodes were grouped into a site if their distance was less than 1.6 Å.

The site coordination number for a given possible Na site was calculated as the number of neighboring anions (i.e. O$^{2-}$, S$^{2-}$, and Cl$^-$) using Voronoi decomposition with solid angle weights as implemented in CrystalNN algorithm[56]. The site volumes were calculated by constructing a convex hull of identified coordinating anions (Supplementary Fig. 5).

## High-throughput screening of Na-ion conductors

Step 1. Basic Materials Check. We firstly filtered all the Na-containing oxide, sulfide and chloride compounds in the ICSD (2017 version), and excluded the compounds with any of following attributes: binary compounds; containing more than one anion species; co-occupancy of Na with other elements; containing anions with disordering or partial occupancy; containing water molecules; having elements with no valence or abnormal valences that have no ionic radii information in the defaulted table of *pymatgen*[55]; the compounds are not charge neutral; containing more than 300 atoms. After this step of basic materials checks, there were a total of 2673 oxides, 153 sulfides, and 80 chlorides for further screening.

Step 2. Percolation radius. Considering the bottleneck size of diffusion channels in the crystal structure, we identify the structures with large percolation radii of $p_r > 0.9$ Å for oxides and $p_r > 0.85$ Å for sulfides and chlorides. The cut-off values were identified based on the key parameters of the crystal frameworks of known Na-ion conductors (Supplementary Table 2).

Step 3. Face-sharing high-CN sites. For a structure to pass the screening, the diffusion network should only consist of high-CN Na sites with CN ≥ 5. These high-CN Na sites should also connect with each

other via face sharing within the cut-off distance to form percolation. The cut-off distance was set to 3.1 Å for oxides and 3.5 Å for sulfides and chlorides.

Step 4. Unique structures. We grouped the compounds with the same crystal structural framework regardless of the cation species using the structure matching algorithm in Pymatgen[55]. In the candidate list (Supplementary Tables 5, 7, and 9), one compound was evaluated to represent a structural framework.

Step 5. Practical consideration. We excluded the compounds containing Au, U, (CO$_3$)$^{2-}$, (SO$_4$)$^{2-}$. We excluded all known Na-SICs and cathode materials, and the Na-Al-O ternary systems. For AIMD simulations, we excluded the compounds with large supercells containing > 300 atoms.

Step 6. AIMD screening. For each crystal structural framework identified in the candidate list, the representative compound was selected to evaluate Na$^+$ diffusion. Aliovalent substitution was performed to change Na content in the compound. For each framework, Na content was changed to have the ratio of the number of Na ions over the total number of identified Na sites to the target range of 0.3–0.7. The energy above the hull for doped composition was calculated, and those with good stability of the energy above the hull <100 meV/atom were further evaluated for Na$^+$ diffusion.

AIMD simulations were first performed at two temperatures, 900 K and 1150 K for oxides and sulfides and 700 K and 900 K for chlorides, as an initial screening following Ref. 43 The materials that melted during AIMD simulations were excluded. For materials with extrapolated Na$^+$ conductivity >0.1 mS/cm at 300 K were further studied by AIMD simulations at more temperatures. The ionic conductivity was evaluated according to the Arrhenius relation,

$$\sigma T = \sigma_0 \exp\left(\frac{-E_a}{k_B T}\right) \qquad (1)$$

where $\sigma$ is conductivity, $T$ is temperature, $E_a$ is the activation energy, $\sigma_0$ is the pre-exponential factor, and $k_B$ is the Boltzmann constant

## Ab initio molecular dynamics simulation

We performed AIMD simulations to study ionic diffusion in supercell models with lattice parameters near or larger than 10 Å. Non-spin mode, a single Γ-centered $k$-point, and a time step of 2 fs were used. In each simulation, the initial temperature was set to 100 K and then the structures were heated to the target temperatures at a constant rate by velocity scaling during a period of 2 ps. All simulations adopted the NVT ensemble with Nosé-Hoover thermostat[57]. The diffusivity $D$ was calculated as the mean square displacement (MSD) over the time interval $\Delta t$,

$$D = \frac{1}{2Nd\Delta t} \sum_{i=1}^{N} \langle [\mathbf{r_i}(t + \Delta t) - \mathbf{r_i}(t)]^2 \rangle_t \qquad (2)$$

where $d$ is the dimension of the diffusion, $N$ is the total number of diffusion ions, $\mathbf{r}_i(t)$ is the displacement of the $i$-th ion at time $t$. The ionic conductivity was calculated according to the Nernst-Einstein relationship,

$$\sigma = \frac{nq^2}{k_B T} D \qquad (3)$$

where $n$ is the mobile ions volume density and $q$ is the ionic charge. Given that the ion hopping is a stochastic process, the statistical deviations of the conductivity were evaluated according to the scheme established in our previous work[58]. The total time duration of AIMD simulations was within the range of 40–400 ps until the total mean-square-displacement reach over 1000–2000 Å$^2$ and the ionic diffusivity converged within a relative standard deviation between 20% and 40% for most data points.

## Experimental methods

**Synthesis.** All preparation processes and sample treatments were carried out in an Ar-filled glovebox ($O_2 < 1$ ppm, $H_2O < 1$ ppm). The family of $Na_{3x}M_{2-x}Cl_6$-contained halide conductors (M = La, Ce, Nd, Sm) were synthesized by ball-milling the starting materials of $LaCl_3$ (Sigma Aldrich, 99.9%), $SmCl_3$ (Sigma Aldrich, 99.9%), $NdCl_3$ (Sigma Aldrich, 99.8%), $CeCl_3$ (Sigma Aldrich, 99.9%), $TaCl_5$ (Sigma Aldrich, 99.9%) and NaCl (Sigma Aldrich, reagent grade) according to the stoichiometric ratios. For ball-milling synthesis, the mixture of precursors was sealed in a zirconia jar (100 mL) under vacuum and was mechanically milled using a plenary high-energy ball-milling machine (PM200, RETSCH) with zirconia balls ($\phi = 5$, 7, and 10 mm, balls/precursors = 40:1 w/w) for 60 cycles. The ball-milling process included 10-min milling and 5-min resting for each cycle. The ball-milling speed was 500 rpm. The as-prepared samples were collected in the glovebox for ionic conductivity measurements.

## Structure characterization

Lab-based X-ray diffraction (XRD) measurements were performed on Bruker AXS D8 Advance with Cu Kα radiation ($\lambda = 1.5406$ Å). Kapton tape was used to cover the sample holder to prevent air exposure. Synchrotron-based powder diffraction patterns were collected using the Brockhouse High Energy Wiggler beamline at the Canadian Light Source (CLS) with a wavelength of 0.3497 and 0.2077 Å. The samples were loaded into 0.8 mm inner diameter polyimide capillaries and sealed with epoxy in an Ar-filled glove box. The X-ray diffraction Rietveld refinement and pair distributed function fittings were conducted by GSAS-2[59] and PDFgui software[60].

## EIS measurements of ionic conductivity

The temperature-dependent ionic conductivities of prepared solid electrolytes were obtained via the EIS measurements of model cells on a multichannel potentiostat 3/Z (German VMP3). The temperature range was between −25 and 55 °C. The applied frequency range was 1 Hz to 7 MHz and the voltage amplitude was 20 mV. The test cell was fabricated as follows: 100–120 mg of the electrolytes were pressed (~300 MPa) into a pellet (diameter: 1 cm, thickness ~0.5–0.7 mm). About 5 mg of acetylene black carbon was then spread onto both sides of the pellet and pressed with ~150 MPa.

## Data availability

All data are provided in the paper and its Supplementary Information. Additional information is available from the corresponding authors upon request. Source data are provided with this paper.

## Code availability

The code used in this study is available from the corresponding author upon request.

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

## Acknowledgements
Y.M. acknowledge the funding support from National Science Foundation Award# 2118838 and the computational facilities from the University of Maryland supercomputing resources. X.S. thanks the funding support from the Natural Sciences and Engineering Research Council of Canada (NSERC), Canada Research Chair Program (CRC), and University of Western Ontario. X.S. also appreciates the help of the BXDS beamline at Canadian Light Source.

## Author contributions
Y.M. oversaw the overall project. Y.M. and S.W. conceived the project. S.W. designed and performed the computation and analyses. J.F., J.L., and S.D. fabricated the samples of the materials and carried out the experiments, characterizations, and data analyses, supervised by X.S. and T.S. S.W. and Y.M. prepared the manuscript with the help of all authors. All authors contributed to the discussions and revisions of the manuscript.

## Competing interests
The authors declare no competing interests.
