## [Peer Review File · Nature Communications]

REVIEWERS' COMMENTS

Reviewer #1 (Remarks to the Author):

a) The authors provide a detailed and acceptable response/discussion on concerted/cooperative mechanisms.

b) Regarding grain boundary effects, the authors list papers on the 'discovery' of known Li-/Na-SICs. However, these studies are largely about 'discovery' of a new material and NOT about 'design principles'.

The authors state their study 'focuses on bulk-phase structure SICs'. I therefore strongly suggest that this should be reflected in the title. e.g. Design principles on bulk-phase ion diffusion in sodium ...

c) The reason for requesting a combined experimental and computational plot is that on page 11, it states: 'in good agreement with the AIMD simulations (Fig. S17).'

I thought that such agreement would be easier to see in a single plot.

Response to Reviewers

Response to Reviewer #1:

a) The authors provide a detailed and acceptable response/discussion on concerted/cooperative mechanisms.

Reply: We thank the reviewer for the comments.

b) Regarding grain boundary effects, the authors list papers on the 'discovery' of known Li-/Na-SICs. However, these studies are largely about 'discovery' of a new material and NOT about 'design principles'. The authors state their study 'focuses on bulk-phase structure SICs'. I therefore strongly suggest that this should be reflected in the title. e.g. Design principles on bulk-phase ion diffusion in sodium ...

Reply: We appreciate the reviewer's suggestion. High ionic conductivity in the bulk phase is a fundamental requirement for a fast ion conductor. Our study marks a critical first step in establishing such design principles, making it a topic of broad interest. Unlike brute-force high-throughput calculation for the discovery of new materials, we focus on elucidating the structures and diffusion mechanisms of Li-ion and Na-ion conductors. This reveals a unique structural feature, face-sharing high-coordination sites, crucial for achieving fast sodium-ion diffusion in solids. Using this feature as the design principle, we identified numerous novel Na-ion conductors in oxides, sulfides, and halides, validating the design principles. Thus, it is justified to keep design principles in the title. In addition, as we explained in the previous comment, the super-ionic conductor is intrinsically a bulk-phase property, as shown in all those previous references. Thus, the term “super-ionic conductor” already implies it is a bulk-phase super-ionic conductor, and adding “bulk-phase” along with “super-ionic conductor” is redundant. For these reasons, the original title is accurate and should be retained.

c) The reason for requesting a combined experimental and computational plot is that on page 11, it states: 'in good agreement with the AIMD simulations (Fig. S17).' I thought that such agreement would be easier to see in a single plot.

Reply: We thank the reviewer for the suggestion. As we explained in the previous response, we put the experimental and computational plots side by side in Supplementary Figure 17.